# Anaesthesia-Induced Transcriptomic Changes in the Context of Renal Ischemia Uncovered by the Use of a Novel Clamping Device

**DOI:** 10.3390/ijms22189840

**Published:** 2021-09-11

**Authors:** Charles Verney, David Legouis, Sandrine Placier, Tiffany Migeon, Philippe Bonnin, David Buob, Juliette Hadchouel, Pierre Galichon

**Affiliations:** 1Common and Rare Kidney Diseases (CoRaKID) Unit, Institut National de la Santé and de la Recherche Médicale (INSERM) U1155, F-75020 Paris, France; charlesverney@gmail.com (C.V.); sandrine.placier@aphp.fr (S.P.); tiffany.migeon@sorbonne-universite.fr (T.M.); david.buob@aphp.fr (D.B.); juliette.hadchouel@inserm.fr (J.H.); 2CoRaKID Unit, Sorbonne Université, F-75020 Paris, France; 3Laboratory of Nephrology, Department of Medicine, University Hospitals of Geneva, 1205 Geneva, Switzerland; David.Legouis@unige.ch; 4Department of Cell Physiology, Faculty of Medicine, University of Geneva, 1205 Geneva, Switzerland; 5Division of Intensive Care, University Hospital of Geneva, 1205 Geneva, Switzerland; 6INSERM U1148 Laboratory for Vascular Translational Science (LVTS) Hôpital Bichat, F-75018 Paris, France; philippe.bonnin@aphp.fr; 7Physiologie Clinique—Explorations Fonctionnelles, Assistance Publique Hôpitaux de Paris (APHP), Lariboisière Hospital, F-75010 Paris, France; 8Department of Anatomopathology, AP-HP, Tenon Hospital, F-75020 Paris, France; 9Surgical and Medical Department of Kidney Transplantation, Assistance Publique–Hôpitaux de Paris (APHP), Pitié-Salpêtrière Hospital, F-75013 Paris, France

**Keywords:** renal ischemia, acute kidney injury, anaesthesia, animal models, surgical clamp

## Abstract

Ischemia is a common cause of acute kidney injury worldwide, frequently occurring in patients undergoing cardiac surgery or admitted to the intensive care unit (ICU). Thus, ischemia-reperfusion injury (IRI) remains one of the main experimental models for the study of kidney diseases. However, the classical technique, based on non-traumatic surgical clamps, suffers from several limitations. It does not allow the induction of multiple episodes of acute kidney injury (AKI) in the same animal, which would be relevant from a human perspective. It also requires a deep and long sedation, raising the question of potential anaesthesia-related biases. We designed a vascular occluding device that can be activated remotely in conscious mice. We first assessed the intensity and the reproducibility of the acute kidney injury induced by this new device. We finally investigated the role played by the anaesthesia in the IRI models at the histological, functional and transcriptomic levels. We showed that this technique allows the rapid induction of renal ischemia in a repeatable and reproducible manner, breaking several classical limitations. In addition, we used its unique specificities to highlight the renal protective effect conferred by the anaesthesia, related to the mitigation of the IRI transcriptomic program.

## 1. Introduction

Acute kidney injury (AKI) is a common condition in the critical care setting, occurring in one over two patients in the intensive care unit (ICU) and associated with both morbidity and mortality [1,2]. Renal ischemia is a frequent cause of AKI [3]. Up to 88% of AKIs are caused, at least in part, by kidney hypoperfusion in critically ill patients [1]. In kidney transplantation, the ischemia is unavoidable during the organ procurement process and has deleterious consequences [4,5].

For these reasons, the ischemia–reperfusion injury (IRI) is a commonly used model of AKI in rodents [6]. The classical procedure to induce an IRI in mice relies on the uni- or bilateral clamping of the renal vascular pedicle(s) with a non-traumatic surgical clamp under general anaesthesia.

However, this technique has several limitations. Firstly, the rapid wear of the clamps, decreasing the severity of the IRI, adds variability to the model. Secondly, it cannot be used to induce multiple occurrences of AKI in the same animal. Iterative episodes of AKI are a common trait of both ICU patients, often subjected to many renal injuries, and kidney grafts, during donor resuscitation and transplantation procedures. As the risk of chronic kidney disease (CKD) increases incrementally with every additional episode of AKI [7], multiple IRIs are also used as a model to trigger the development of CKD [8]. Finally, multiple IRIs in mice could allow the modelling of the ischemic preconditioning [9,10], in which a first and brief episode of ischemia can induce protective pathways.

Lastly, the classical procedure must be performed in sedated animals. Anaesthesia can modify the body temperature [11,12] and the renal blood flow, two major parameters influencing AKI severity [13,14] and different anaesthetics lead to different outcomes [13]. In the same line, recent evidence suggests that anaesthesia by itself may impact the AKI lesions in response to the IRI [15].

Therefore, the aim of our work was to design a device that allows (1) the induction of repeated episodes of renal ischemia (2) in conscious animals. For this purpose, we used segments of flexible silicone tubing mounted on a simple woven thread with the extremities maintained at the surface of the mouse’s skin. We named this device the RIRI clamp, for ‘repeated ischemia-reperfusion injury’ clamp. We showed that the renal injury induced by the RIRI clamp was more severe when animals were conscious. We discussed the possible inferences of body temperature, stress and metabolism in light of the genome-wide transcriptomic data comparing the ischemic kidneys of anaesthetised and conscious mice. Altogether, we uncovered the consequences of anaesthesia on the course of renal ischemic injury, paving the road for future discoveries in the field of perioperative nephroprotection.

## 2. Results

### 2.1. Absence of a Commercially Available Device Suitable to Induce Multiple Episodes of IRI in Mice

Only a few devices, other than the routinely used non-traumatic surgical clamp, are available. To our knowledge, the only one that allows to perform repeated occlusions of the renal vascular pedicle in mice is the vascular occluder developed by Fine Scientific Tools (FST). If multiple efficient IRIs could indeed be performed with this device, we also observed that the simple insertion of this inflatable clamp could damage the renal vascular pedicle, thereby inducing acute tubular necrosis (Appendix A). We concluded that, although such a tool might be used to perform a renal IRI in larger animals, such as rats, it is not suitable for mice.

### 2.2. Generation of New Device Allowing Repeated Renal Ischemia in Conscious Mice

Therefore, we set out to design a novel device, which we named the RIRI clamp, for repeated ischemia-reperfusion injury clamp. In Figure 1 and explained in detail in the Materials and Methods section, we designed a tunnelled clamp with remote access allowing us to perform delayed, non-invasive renal ischemia-reperfusion in conscious mice. Briefly, two flexible cylinders mounted on a thread are positioned around the renal pedicle (Figure 1A). The thread is inserted within a 16 G-catheter placed under the skin, in the dorsal position, with its two extremities hanging outside of the mouse. Pulling of the two extremities of the thread results in the clamping of the pedicle by the two cylinders (Figure 1B). Releasing the tension of the thread allows the reperfusion. A trichrome Masson staining allowed us to show that the RIRI clamp insertion surgery did not induce any lesion in the kidney, up until 48 h after the surgery (Appendix A).

### 2.3. Efficient IRI Using the RIRI Clamp

To ensure that our device efficiently interrupted the renal blood flow, we first checked during the surgery, after the positioning of the clamp and before the abdominal wall was sutured, that the clamped kidney became cyanotic when the thread was pulled and regained its original colour when the clamp was put back in the open position, as with the classical IRI method. We consistently observed the expected changes in all of the mice (*n* = 24, Appendix A). Since our device is designed to perform the ischemia at least 24 h after the surgery in conscious mice, we then used ultrasound Doppler imaging to characterize the hemodynamic changes in the intrarenal arteries in three slightly anesthetised mice, using isoflurane (Figure 2). When the clamp was open, a normal renal blood flow was observed (Figure 2A). As soon as the clamp was tightened, the renal blood flow was interrupted immediately (Figure 2B) and was then reinstated when the clamp was loosened (Figure 2C).

We then studied the structure in mice subjected to a 20-min renal ischemia, performed 24 h after the positioning of the clamp, with a reperfusion time of 18 h to 42 h. Of the mice subjected to this procedure, 97% presented typical ischemia-induced tubular modifications, i.e., loss of the brush border, flattening of the tubular epithelium and intratubular casts (Figure 2D). In order to precisely observe the renal injury imposed by the open RIRI clamp and the ischemia when it is tightened, we measured the transcriptional expression of kidney injury molecule 1 (KIM1) (Kidney Injury Molecule 1) by quantitative reverse transcription polymerase chain reaction (RT-qPCR). KIM1 is a type 1 transmembrane protein domain, the expression of which in the proximal tubule is highly increased upon aggression. The KIM1 mRNA level strongly increased 18 h after a 20-min ischemia, and then progressively decreased between 24 and 42 h after the ischemia (Figure 2E), thus showing that the clamp did not impair the capacity of the renal tubular epithelium to recover.

### 2.4. The RIRI Clamp Can Be Used to Induce Ischemia in a Large Number of Mice in a Short Period of Time

Studies in rodents have shown that the time of day at which an ischemia is triggered can influence the severity of a myocardial infarction [16,17,18]. Indeed, the infarct size in hearts subjected to ischemia at the sleep-to-wake transition was significantly increased compared to hearts subjected to ischemia at the wake-to-sleep transition. The mechanisms underlying this time-of-day susceptibility to an ischemic injury include the circadian modulation of hypoxic signalling and inflammatory pathways [19,20]. Therefore, it can be expected that this phenomenon also occurs in the context of renal ischemia, although it has not been demonstrated yet. In our experience, it was difficult to perform renal ischemia in more than 16 mice in half a day. Using the RIRI clamp, we were able to perform a 20 min ischemia on 12 mice in only 40 min. This may prevent the bias related to the circadian rhythm or to other imperceptible factors varying over time by inducing all the IRI lesions in a short period of time, even on a large number of mice.

### 2.5. Multiple Episodes of IRI Can Be Induced with the RIRI Clamp

We originally designed the RIRI clamp in order to trigger multiple IRIs in the same animal, thus allowing the generation of a more relevant clinical model. Indeed, as mentioned in the introduction, ICU or transplant patients often suffer from multiple AKI episodes and this repetition significantly increases the risk of developing chronic kidney disease.

We first tested the feasibility of inducing repeated ischemic episodes on an anaesthetised mouse in order to verify the efficacy of the clamping visually, through the changes in colour of the kidney. The vascular pedicle was clamped six times for 5 min, each separated by 5 min. The kidney became cyanotic when the clamp was tightened and regained its original colour when the clamp was released during each procedure.

We then assessed the efficacy of repeated vascular occlusions by ultrasound imaging, 24 h after surgery. As shown in Figure 3 and Video S1, the blood flow was interrupted during the first 26-min ischemia (Figure 3A–C), restored (Figure 3D) and then interrupted again during a second 5-min ischemia (Figure 3E,F), performed 10 min after the first one.

### 2.6. IRI Induced in Conscious Animals Is More Severe Than in Anaesthetised Ones

We set up to compare the efficacy of the RIRI clamp to that of the gold standard, i.e., the classical non-traumatic clamp. For that purpose, the contralateral kidney was excised in order to assess the consequences of the ischemia-reperfusion on renal function. In order to have the exact same conditions, a 30 min ischemia under anaesthesia was performed immediately after the nephrectomy with either the classical clamp (group ‘Classic 30’’) or the RIRI clamp (group ‘RIRI 30’ anaesthetised’). As shown in Figure 4A,B, the plasma creatinine and urea concentration were similar after 24 h of reperfusion between the two clamps. Therefore, the RIRI clamp is as efficient and reproducible as the classic one to induce ischemic AKI in anaesthetised animals. We then evaluated the consequences on the renal function of a 30-min ischemia with the RIRI clamp in conscious animals, 24 h after the contralateral nephrectomy and positioning of the clamp (group ‘RIRI 30’ conscious’). The plasma urea and creatinine levels were significantly higher in the conscious animals than in the anaesthetised animals (Figure 4A,B). Similarly, there was no difference in the histological changes after a 30-min ischemia performed with either the classic or RIRI clamp in anaesthetised animals, whereas the lesions induced by a 30-min ischemia in conscious animals were significantly more severe (Figure 4C–I).

We then tested different durations of ischemia (5, 10 and 20 min) in order to find the one required to obtain an alteration of the renal structure and function similar to the one obtained with a 30-min ischemia with the classical atraumatic clamp. As shown in Figure 5A, KIM1 mRNA expression increased after 5 and 10 min of ischemia, compared with the non-ischemic kidney, and then decreased. This was probably due to the important cell death caused by the 15–30 min of ischemia, observed on the periodic acid–Schiff-stained sections (Figure 5B and Figure 6A–D). The plasma urea and creatinine concentrations were correlated to the ischemia duration between 5 and 15 min and then reached a plateau (Figure 5C,D). The values after 10 min were comparable to the ones obtained after a 30-min ischemia performed under anaesthesia. Taken together, these observations clearly demonstrate that the consequences of an ischemia performed in a conscious animal are more severe than those of an ischemia performed under anaesthesia.

It is well known that anaesthesia decreases the body temperature [11,12]. It is particularly true in small animals, for which the relatively important body surface area promotes heat loss and hypothermia. The protective role of hypothermia during renal ischemia has been well documented in rats [21,22] and very recently in mice [23], and also in kidney donors [24]. Therefore, we hypothesised that the increased severity of the conscious RIRI ischemia could be due to an absence of or moderate decrease in body temperature.

We measured the body temperature in the following groups: (1) ischemia induced by the classical clamp in two anaesthetised mice (with xylazine and ketamine), and (2) ischemia induced by the RIRI clamp in two conscious mice (with a brief anaesthesia with isoflurane to tighten the clamp). The mice of the first group were maintained on a heating pad at 37.5 °C throughout the procedure. During the clamping period, they were covered with a survival blanket and a 40-watt lamp was placed 20 cm above the animals. The mice of the second group were kept in a contention tube to allow the monitoring of body temperature, and a 40-watt lamp was placed 20 cm above the animals. The body temperature was monitored with a rectal probe in each group and recorded every minute. After the anaesthesia, all of the mice experienced a modest decrease in body temperature, but the recovery of the baseline temperature was better in the RIRI group when the mice awoke (Appendix A). This last result highlights the inability of anaesthetised mice to autoregulate their body temperature.

### 2.7. Characterisation of the Protective Effect of Anaesthesia at the Transcriptomic Level

#### 2.7.1. Anaesthesia Protects from IRI Independently of Hypothermia

We then tested whether or not the anaesthesia by itself, independent of the decreased body temperature, could have a protective effect on the functional and histological consequences of renal ischemia. We performed a 20-min ischemia with the RIRI clamp in six mice anaesthetised with ketamine and xylazine and six conscious mice. Both groups were maintained in a neonatal incubator set at 34 °C during the ischemia, in order to maintain the body temperature. As shown in Figure 7A, the conscious group exhibited a significantly higher plasma creatinine level than the anaesthetised group 24 h after the ischemia. The histological lesions tend to be more severe in the conscious group (Figure 7B). These observations suggest that the anaesthesia improves the outcome of renal ischemia independently of hypothermia.

#### 2.7.2. Transcriptional Changes Induced by the Anaesthesia

In order to further characterise the effects of anaesthesia, we compared the transcriptomic profile, generated by RNA-sequencing, of control kidneys and ischemic kidneys from anaesthetised and conscious animals (*n* = 4 in each group). Out of 11,660 transcripts, 2637 (23%) were significantly (false discovery rate (FDR) < 0.05) regulated by ischemia (ischemia reactive genes). A logistic regression pathway analysis identified 323 Gene Ontology Biological Pathways (GO_BP) enriched with the corresponding genes. These GO_BP are summarised in the following figure using the Reduce + Visualize Gene Ontology (REVIGO) tool, highlighting the importance of monocarboxylic acid metabolism regulation in renal ischemia-reperfusion injury (Appendix A). Thirty-five Kyoto Encyclopaedia of Genes and Genome (KEGG) pathways were significantly enriched in ischemia-reperfusion-regulated genes, with a majority of metabolic pathways (Appendix A). The directional analysis (Appendix A) identified 48 pathways enriched in genes with increased expression after ischemia-reperfusion (including pathways related to cell cycle and epithelial differentiation), and 39 pathways enriched in genes with decreased expression after ischemia-reperfusion (mostly metabolic pathways). We created a ‘renal ischemia-reperfusion’ gene set with the 2637 genes regulated by ischemia. After accounting for multiple testing, we found only one gene to be significantly regulated by anaesthesia during ischemia-reperfusion injury: slc5a12 (sodium/glucose cotransporter, 71-fold decrease in ischemia with anaesthesia compared to conscious ischemia, FDR = 0.03). This gene was also downregulated by ischemia compared to sham surgery (30-fold, FDR = 4.7 × 10^−4^). However, the more holistic approach of logistic regression pathway analysis on genes regulated by anaesthesia during renal ischemia-reperfusion injury (Appendix A) identified 35 GO_BP pathways and 4 KEGG pathways as significantly enriched in genes regulated by anaesthesia during ischemia-reperfusion. Many of these pathways were again metabolic pathways, suggesting epistasis between anaesthesia and ischemia-reperfusion. Among all of the tested gene sets, the ‘renal ischemia-reperfusion’ gene set was the most enriched (OR = 1.5, FDR = 1.64 × 10^−10^), showing a strong interaction between ischemia and anaesthesia at the transcriptomic level. As shown in Figure 8, we found that most of the genes downregulated by ischemia in conscious mice were not downregulated in anaesthetised mice, whereas anaesthesia had no effect on genes upregulated by ischemia. Finally, the expression of another group of genes was downregulated by the anaesthesia but unaffected by the ischemia (Figure 8 and Appendix A).

## 3. Discussion

Numerous previous studies reported the importance of experimental variables in modifying the consequences of ischemia-reperfusion on kidney injury. Body temperature [12], volume management [25], the circadian rhythm (10.1172/JCI80590) or the nutritional status [26] are such modifiers. This is both a limitation of the experimental model and an opportunity to identify therapeutic targets. The technique that we describe here allows controlling the main previously described modifiers of ischemia-reperfusion injury by providing a stable body temperature, the absence of surgical complications and a precise duration for ischemia-reperfusion. Performing renal ischemia-reperfusion injury in conscious mice provided a surgery-free procedure with a stable, physiological body temperature, and the possibility to perform the same procedure in multiple mice at the same time. Furthermore, this technique allowed the study of the effects of anaesthesia separately, as a potential modifier of ischemia-reperfusion injury. We found that anaesthesia significantly mitigates renal ischemia-reperfusion injury, by modulating, at least in part, the transcriptomic changes induced by the ischemia. These findings have clinical implications, as general anaesthesia is an option in many clinical situations at risk of AKI (surgery, intensive care), although it is not mandatory in many cases [27,28].

This technique can be used to address other questions: is the type and the dosage of anaesthetic drugs important to alter the course of ischemic acute kidney injury? What is the consequence of repeating ischemic insults at various intervals? The effect of multiple ischemia within the same procedure was extensively studied in preconditioning and postconditioning experiments, showing a strong protective effect in animal models [29]. Repeated ischemias after recovery (with an interval of weeks or months) were performed in studies evaluating the long-term effect of ischemia [8,30]. However, performing repeated ischemias during the recovery phase (within the first days) after a first hit was technically not possible because of (1) the frailty of the recovering animals and their sensitivity towards a second anaesthesia at an early time point and (2) the scarring process around the renal vascular pedicle induced by the first clamping. Our technique allows studying repeated hits during this phase of recovery, which is most relevant to the clinical setting of AKI in ICU patients, where multiple hits most often occur within the first few days when patients are unstable and submitted to intensive treatments.

Studying the consequences of renal ischemia-reperfusion injury with and without anaesthesia in our model also allowed us to gain insights on the underlying mechanisms of kidney injury and resilience. We found that the downregulation of metabolic pathways is a hallmark of the transcriptomic changes induced by ischemia. Importantly, the anaesthesia counteracts this effect, but does not prevent the expression of genes upregulated after ischemia. This, combined with the observation that anaesthesia is protective, suggests that genes upregulated in the kidney after ischemia are not essential for the pathogenesis of acute kidney injury, but might reflect an adaptative process in response to injury that is not blunted by anaesthesia. Some of these genes upregulated by ischemia are indeed known to be protective at the early stage of acute kidney injury, such as KIM1 [31]. On the other hand, the prevention by anaesthesia of gene downregulation after ischemia suggests that the downregulation is directly associated to injury, either as a marker or as a cause of injury. Further research is needed to determine if the genes downregulated by the anaesthesia could be protective against ischemic injury, or if they are “bystander” genes with no effect on injury. In conclusion, the RIRI clamp represents a major technical advance for the study of renal ischemia-reperfusion injury, allowing to explore new variables modifying the severity of ischemic kidney injury, and delineating the underlying transcriptomic changes in the kidney.

## 4. Materials and Methods

### 4.1. Animals

We used 8- to 15-week-old (24–30 g) C57Bl/6J males from Charles River Laboratories (Saint-Germain-sur-l’Arbresle, France). They were maintained under routine vivarium conditions, in a pathogen-free animal facility, in ventilated cages with a 12/12 photoperiod, at 22 ± 2 °C with a 55 ± 10% humidity. Food and water were provided ad libitum.

### 4.2. Testing of the Vascular Occluder Inflatable Device (InterFocus^®^ Ref. 18080-02)

The mouse was anaesthetised with a mixture of ketamine (100 mg/kg) and xylazine (10 mg/kg), diluted in sterile saline to a final volume of 100 µL/10 g body weight, administered intraperitoneally. Following induction of anaesthesia, which usually takes 5 min, the left side of the abdomen was shaved and disinfected with Betadine^®^ solution. The animal was kept on a heating pad, set at 37 °C, during the whole procedure (25 min). A posterior subcostal incision was made in the left side and the left kidney was exteriorised. The renal pedicle was then dissected. The occluder cuff was wrapped around the exposed pedicle (renal artery and vein) and secured in place using silk suture thread passed through the eyelets and tied securely. The actuating tube was placed under the abdominal skin and exteriorised through an incision anterior to the left hindlimb. A Vicryl 4-0 suture (Ethicon) was used to first close the muscle layer, followed by the closing of the skin. At the end of the procedure, the animal received an intraperitoneal injection of buprenorphine (0.05 mg/kg, diluted in sterile saline to a final volume of 100 µL/10 g body weight).

### 4.3. Nephrectomy and Positioning of the RIRI Clamp

The following material was used to build the RIRI clamp: a 16 G (grey) catheter, woven thread and silicone tubing (0.030 × 0.065 in, Phymep #807000). Two cylinders of 4-mm-long tubing were mounted on a 6-cm-long thread (Figure 1). The flexible portion of the catheter was cut at 1–1.5 cm from the solid part.

The mouse was anaesthetised with an intraperitoneal injection of a mix containing ketamine and xylazine (100 mg/kg and 10 mg/kg, respectively). Following the induction of anaesthesia, the left (for the RIRI positioning) and right (for nephrectomy) sides of the abdomen were shaved and disinfected with Betadine^®^ solution. The animal was kept on a heating pad, set at 37 °C, during the whole procedure (25 min without nephrectomy, 30 min with nephrectomy).

For the nephrectomy, a 1-centimetre incision was performed on the right flank of the animal, posterior to the costal margin and perpendicular to the spine. The right kidney was exteriorised. Following the ligature of the vascular pedicle using braided silk (Silk suture 5/0, FST), the kidney was excised. A Vicryl 4-0 suture was used to close the muscle layer and the skin. The animal was then placed on its right side to perform the surgery, allowing the positioning of the RIRI clamp. A 1-centimetre incision was performed on the left flank, posterior to the costal margin and perpendicular to the spine. Another 1-centimetre incision was performed anterior to the left hip, perpendicular to the spine and the skin was gently separated from the muscle layer using tissue separating scissors to allow the insertion of the catheter.

The kidney was exteriorised through the first incision. The renal vascular pedicle was carefully dissected with fine-point tweezers to remove the perihilar adipose tissue, exposing the blood vessels. The thread of the RIRI clamp was placed around the pedicle with the two cylinders on each side of the pedicle. The kidney was then released into the abdominal cavity. The correct position of the silicone cylinders around the vascular pedicle was then checked.

The catheter was inserted into the posterior incision with the flexible end towards the kidney. The extremities of the thread were then inserted within the tube of the catheter and exposed through the other end. The muscle layer and skin above the kidney were closed with a Vicryl 4-0 suture. The skin of the second incision was closed and the solid portion of the catheter was fixed to the skin of the mouse with a Vicryl 4-0 suture. After completion of the surgery, the mouse received a subcutaneous injection of buprenorphine (0.05 mg/kg) and was maintained in a neonatal incubator set at 32 °C until full recovery from the anaesthesia and then at 28 °C. The following day, the mouse was weighed and received a second injection of buprenorphine before inducing the ischemia.

### 4.4. Induction of Ischemia with the RIRI Device

At least 24 h after the clamp-positioning surgery, the mouse was lightly sedated by exposure to 2% isoflurane for 1 min on a heating pad. Once the mouse was anaesthetised, the cap of the catheter was removed, and the clamp was closed by pulling the external extremities of the thread. Once the clamp was in tension, the cap was closed, and the mouse was placed in its cage in the neonatal incubator at 34 °C to prevent any decrease in body temperature following the sedation. One minute before the end of the ischemia, the mouse was again sedated by exposure to isoflurane for 1 min. The catheter cap was removed to release the tension of the thread and then closed again. The mouse was kept in the neonatal incubator at 28 °C until the end of the experimental procedure.

### 4.5. Renal Ischemia-Reperfusion Injury Using the Non-Traumatic Surgical Clamp

Following the excision of the right kidney (see above), the ischemia of the left kidney was performed as described in [8].

### 4.6. Non-Invasive Ultrasound Assessment of Renal Hemodynamic

Ultrasound examination was carried out under isoflurane anaesthesia using an echocardiograph (Acuson S3000; Siemens, Erlangen, Germany) equipped with a 14 MHz linear transducer (14L5 SP). Mice were placed on a heating blanket (38 °C) to avoid hypothermia. A cross-sectional view of the abdomen, parallel to the spine allowed visualisation of the left kidney, the aorta, the inferior vena cave and the right kidney. A colour-coded Doppler helped to recognise the left renal artery between the aorta and the left kidney, as well as in the right renal artery. A pulsed Doppler sample was placed on the longitudinal axis of the left renal artery and the spectral analysis of the velocity waveform was recorded. Angle correction between the axis of the Doppler beam and the longitudinal axis of the studied artery permitted to record the true values of the blood flow velocities.

### 4.7. Tissue Collection

At the end of the procedure, mice were anaesthetised with an intraperitoneal injection of a mix containing ketamine and xylazine (100 mg/kg and 10 mg/kg, respectively). From the retroorbital sinus, 500 µL of blood was collected. The kidneys were then collected. The median part of each kidney was fixed in Formalin–Acetic acid–Alcohol (FAA) overnight and embedded in paraffin for histological examination. The remaining parts were frozen in liquid nitrogen for RNA and protein analysis.

### 4.8. Histological Analysis

For histological evaluation, 3-micrometre sections were stained with Masson’s trichrome or periodic acid–Schiff. The scoring was performed in a masked manner on coded slides by two different investigators, using the following scale: 0: no tubular damage; 1: damage in 1–20% of the corticomedullary tubules; 2: damage in 20–100% of the corticomedullary tubules; 3: same as 2 plus cortical dilatation; 4: same as in 2 plus cortical oedema and zones of infarction.

### 4.9. RNA Extraction and RT-qPCR

The total RNA was extracted from the mouse kidneys using Tri-reagent and BAN (Molecular Research Center Inc.—Euromedex, Souffelweyersheim, France). The RNA quality was checked by control of optical density at 260 and 280 nm. The contaminating genomic DNA was removed by RNase-free DNAse (Thermo Fisher Scientific, Illkirch, France) for 30 min at 37 °C. The cDNA was synthesised from 1 µg of purified RNA using Maxima Enzyme Mix (Thermo Fisher Scientific, Illkirch, France) according to the manufacturer’s instructions. Real-time PCR amplification was performed using the LightCycler^®^ 480 Instrument and SYBR Green I Master (Roche LifeScience, Meylan, France). All of the samples were assayed in duplicate, and the average value of the duplicate was used for quantification. The analysis of the relative gene expression (RGE) was performed using the Pfaffl equation [32]. The geometric mean of the E^∆Ct^ of 2–3 housekeeping genes (*GAPDH* and *HPRT* for Figure 2, *Gusb*, *Ubc* and *RPL32* for Figure 5) was used as the housekeeping reference. The sequences of primers used in our studies are listed in Appendix A.

### 4.10. Statistical Analysis

The statistical test used for the analysis is indicated in the Figure legends. A difference between groups was considered significant when *p* < 0.05. The statistical analysis was performed using the GraphPad PRISM software v6 (Ritme, Paris, France).

### 4.11. RNA Extraction and Bulk RNA Sequencing

For bulk RNA sequencing, 12 kidneys corresponding to 3 conditions: nephrectomy as normal control (group 1, *n* = 4), renal ischemia in conscious mice followed by 24 h reperfusion (group 2, *n* = 4) and renal ischemia in anesthetised mice followed by 24 h reperfusion (group 3, *n* = 4) were included.

The total RNA was extracted from kidneys with Trizol reagent (Invitrogen, Waltham, MA, USA) according to the manufacturer’s instruction. The RNA quantification was performed with a Qubit fluorimeter (Thermo Fisher Scientific, Illkirch, France) and the RNA integrity was assessed with a Bioanalyzer (Agilent Technologies, Les Ulis, France). The TruSeq mRNA stranded kit from Illumina was used for library preparation with 700 ng of total RNA as input. The library molarity and quality were assessed with the Qubit and Tapestation using a DNA High sensitivity chip (Agilent Technologies). The libraries were pooled at 2 nM and loaded for clustering on a Single-read Illumina Flow cell for an average of 25 million reads per sample. Reads of 100 bases were generated using the TruSeq SBS chemistry on an Illumina HiSeq 4000 sequencer (Illumina, Evry, France). After the quality control was performed with FastQC, the reads were mapped to the mm10 genome using STAR software (http://star.mit.edu, accessed on 1 June 2021). Tables of counts were thus generated with HTSeq. The normalisation and differential expression analysis was performed with the R package edgeR, for the genes annotated in the reference genome. We filtered out very lowly expressed genes and normalised data according to the library size (sequencing depth) and the RNA composition (dispersion). The differentially expressed genes were estimated using a general linear model (GLM) approach, negative binomial distribution and a quasi-likelihood F-test.

We identified 2637 transcripts corresponding to 2531 National Center for Biotechnology Information (NCBI)-referenced genes that were regulated by ischemia, by comparing group 2 to group 1 using a bilateral *t*-test with the FDR cutoff for significance set at 5%. We created a “renal ischemia-reperfusion” gene set with these genes significantly regulated by ischemia. In a secondary analysis, we used a logistic regression pathway analysis (LrPath) in order to identify other known gene sets enriched with ischemia-regulated genes, and used the REVIGO (Reduce + Visualize Gene Ontology) tool to provide a graphical representation of these results.

Next, we studied the effect of anaesthesia during renal ischemia on gene regulation by comparing the gene expression in group 3 vs. group 2, using a bilateral *t*-test. We used a logistic regression pathway analysis (LrPath) in order to identify previously described gene sets (including our newly defined “renal ischemia reperfusion” gene set) enriched in genes modified by anaesthesia during renal ischemia-reperfusion. We used the REVIGO tool to provide a graphical representation of these results.

## Figures and Tables

**Figure 1 ijms-22-09840-f001:**
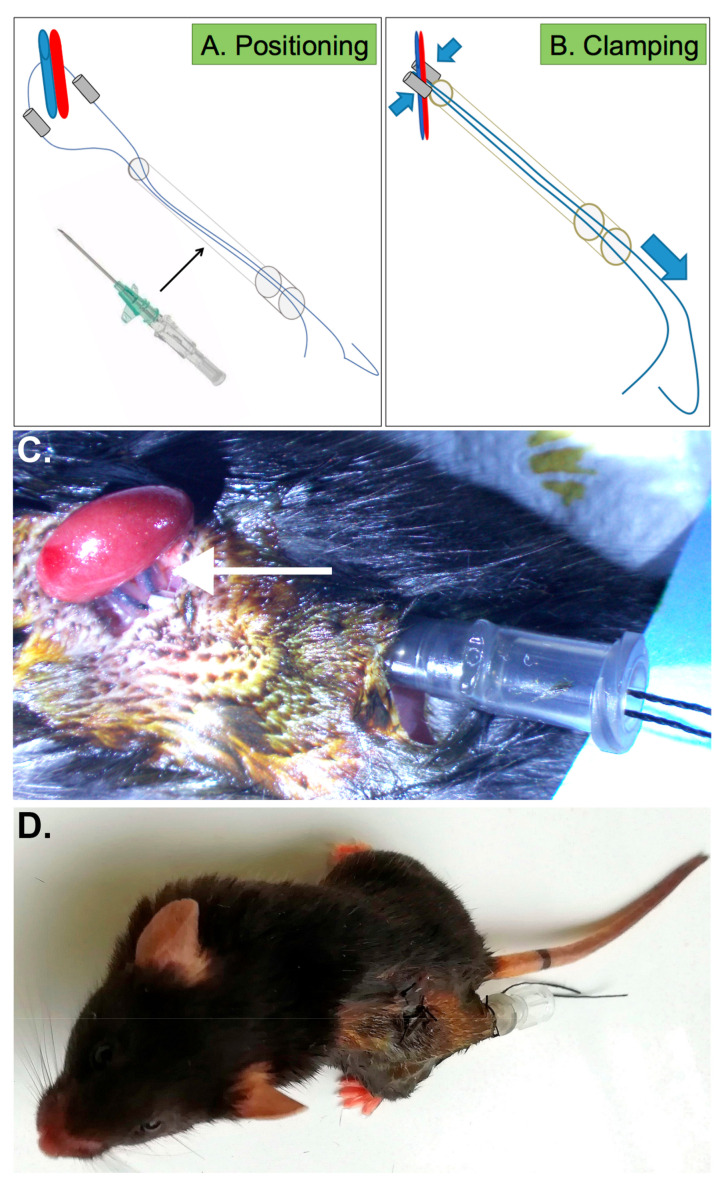
The repeated ischemia-reperfusion injury (RIRI) clamping device. (**A**,**B**) Schematic representation of the RIRI clamping device in the open (**A**) and closed (**B**) positions. (**C**) Representative image of the thread/silicone tubings positioned around the renal vascular pedicle (arrow), the extremities of which are inserted into a 16 G (grey) catheter. (**D**) A conscious mouse with the RIRI clamping device.

**Figure 2 ijms-22-09840-f002:**
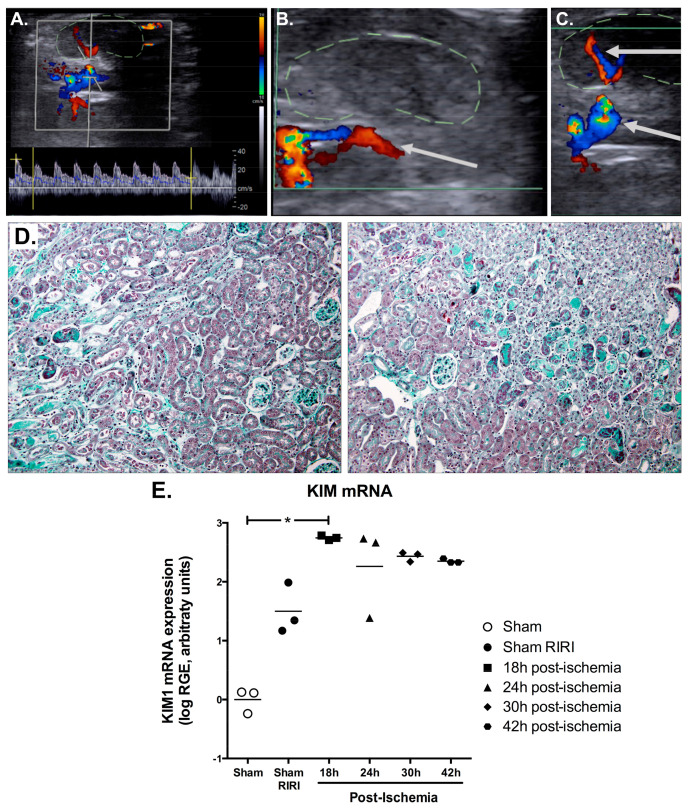
Control of the efficacy of the RIRI clamp. (**A**–**C**) The renal blood flow was monitored using ultrasound Doppler imaging in slightly anaesthetised mice following a cross-sectional view of the abdomen parallel to the spine, from top to bottom on the image: left kidney, left renal artery, aorta, inferior vena cava and right renal artery. When the clamp was in the open position (**A**), blood flow could be observed in the renal vascular pedicle. This flow was interrupted when the clamp was tightened (**B**) and then re-established when the clamp was released (**C**). (**D**) Representative images of mouse kidneys subjected to a 20-min ischemia followed by 18 h of reperfusion and stained with Masson’s trichrome. A loss of the brush border of the proximal tubules, flattening of the tubular epithelium and intratubular casts at the corticomedullary junction were observed. The tubular damages were confirmed by an increased level of kidney injury molecule 1 (KIM1) transcripts (**E**), which then gradually decreased between 24 and 42 h of reperfusion. The individual log values of KIM1 relative gene expression (RGE) are plotted and the bars correspond to the average ± s.e.m. * *p* < 0.05 vs. Sham kidneys. (Kruskal–Wallis one-way analysis of variance followed by Dunn’s multiple comparisons tests.).

**Figure 3 ijms-22-09840-f003:**
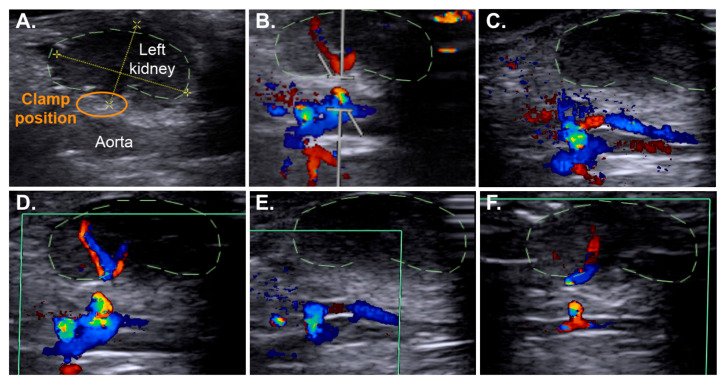
Multiple ischemic episodes can be performed with the RIRI clamp. The use of ultrasound Doppler imaging allowed the observation of the successful multiple interruptions of the renal blood flow using the RIRI clamp in a slightly anaesthetised mouse. (**A**) Position of the left kidney. (**B**) Recording of the blood flow velocity waveforms in the left renal artery before the first clamping. The red and blue colours correspond to the blood flowing away or towards the probe, respectively. Following the pulling and tightening of the thread of the RIRI clamp, the renal blood flow is interrupted (**C**,**E**), although it persists in the right renal artery (blue colour) on the opposite side of the aorta, and is then re-established when the thread was released (**D**,**F**).

**Figure 4 ijms-22-09840-f004:**
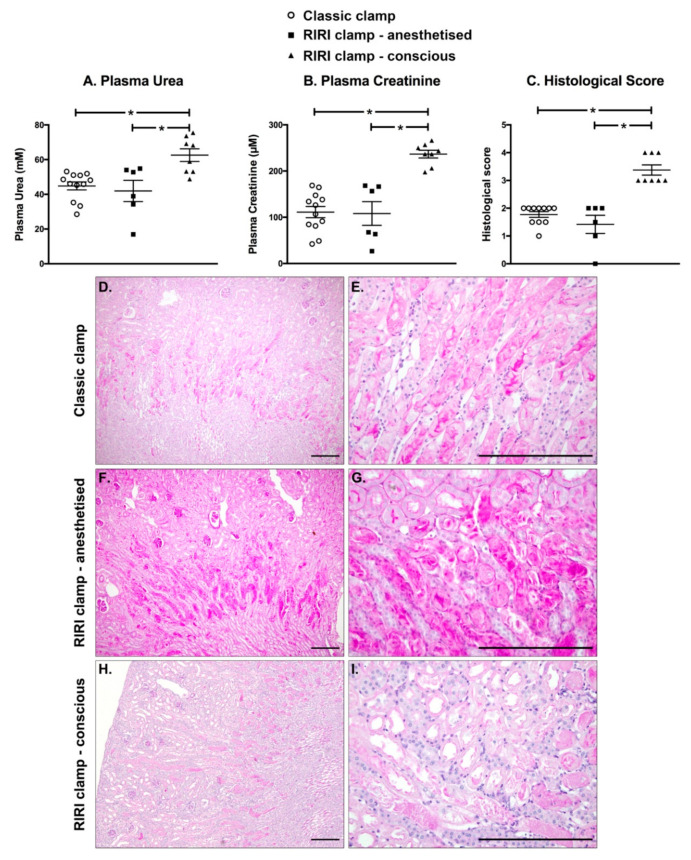
A renal ischemia in conscious mice caused a more pronounced alteration of the renal structure and function Table. 30 min ischemia in anaesthetised mice, as evidenced by a similar increase in the level of the plasma concentration of urea (**A**) and creatinine (**B**) and histological score (**C**), based on the observation of kidney sections stained with Schiff’s periodic acid (**D**–**I**; scale bar: 200 µm). However, the same duration of renal ischemia in conscious mice induced a more severe acute kidney injury than in anaesthetised animals, with cortical dilations (**H**,**I**) in addition to the lesions observed at the corticomedullary junction (**E**–**G**). Individual values are plotted, and the bars correspond to the average ± s.e.m. * *p* < 0.05 between the two groups is indicated by the line (ordinary one-way analysis of variance followed by Sidak’s multiple comparisons tests).

**Figure 5 ijms-22-09840-f005:**
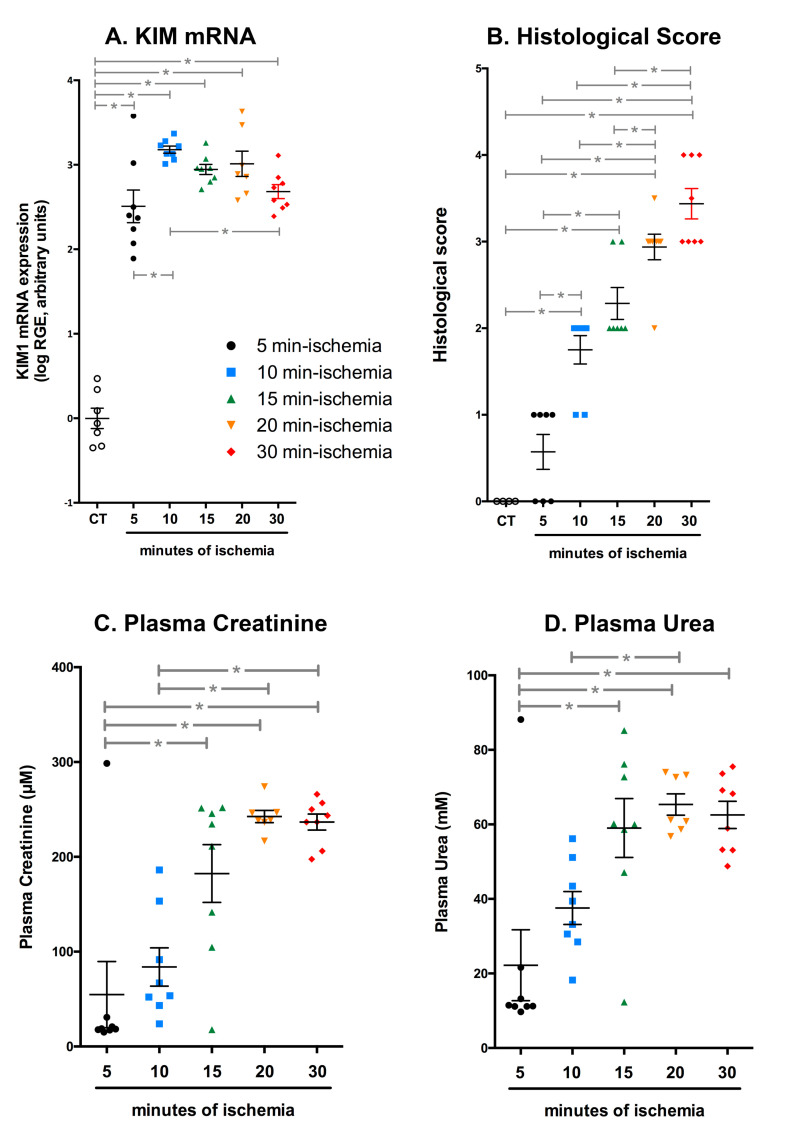
Ten minutes of ischemia with the RIRI clamp in conscious animals induce a similar alteration of the renal structure and function than a 30-min ischemia in anaesthetised mice. (**A**) Quantification by quantitative reverse transcription polymerase chain reaction (RT-qPCR) of the level of KIM1 transcripts in the kidney of mice subjected to 5 to 30 min of ischemia. (**B**) Evaluation of the histological score, based on the observation of kidney sections stained with Schiff’s periodic acid (see Figure 6), showed that a 10 min ischemia with the RIRI clamp in conscious animals induced a similar degree of acute tubular injury than a 30-min ischemia in anaesthetised animals (compare with Figure 4C). The same conclusion was drawn from the measurement of the plasma concentration of creatinine (**C**) and urea (**D**). All these parameters were measured 24 h after the ischemia. The values are plotted, and the bars correspond to the average ± s.e.m. * *p* < 0.05 between the two groups indicated by the line (ordinary one-way analysis of variance followed by Sidak’s multiple comparisons tests).

**Figure 6 ijms-22-09840-f006:**
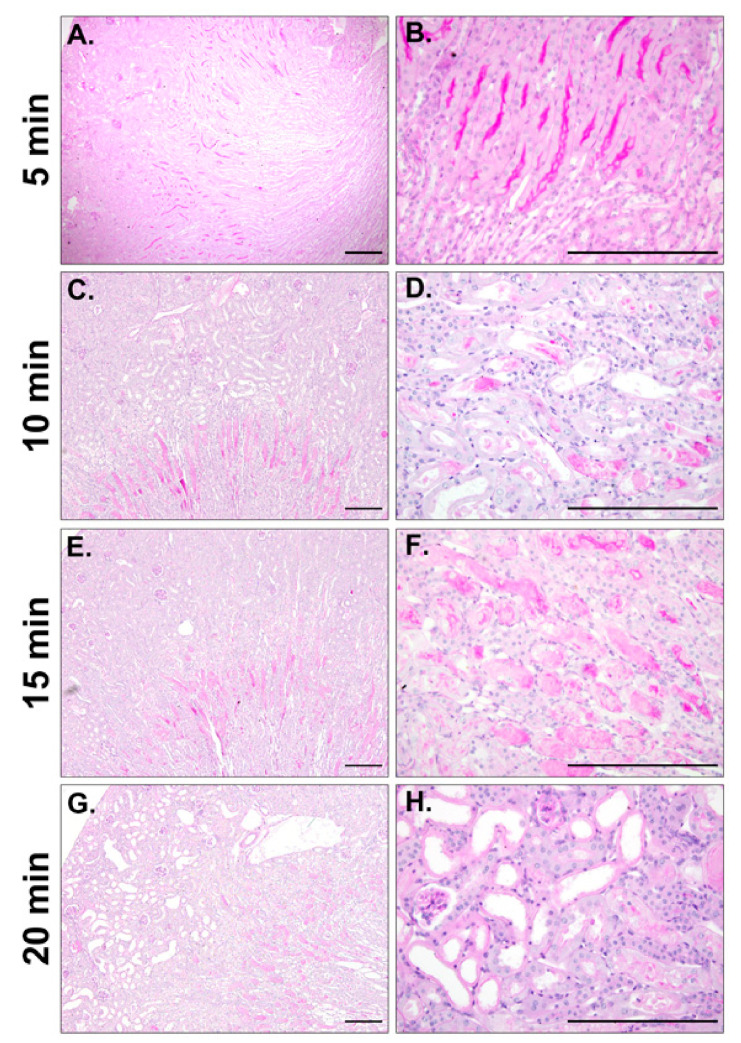
Representative images of the tubular modifications obtained after 5 to 20 min of ischemia. The kidneys were collected 24 h after ischemia and stained with Schiff’s periodic acid staining. A 5-min ischemia (**A**,**B**) induced no visible alteration of renal histology: the brush border was still visible in all the proximal tubules and no apoptotic cells or casts were observed. After a 10-min (**C**,**D**) or 15-min ischemia (**E**,**F**), a loss of the brush border, a thinning of the epithelium and casts were observed at the corticomedullary junction. After 20 min of ischemia (**G**,**H**), additional cortical dilations were seen. Scale bar: 200 µm.

**Figure 7 ijms-22-09840-f007:**
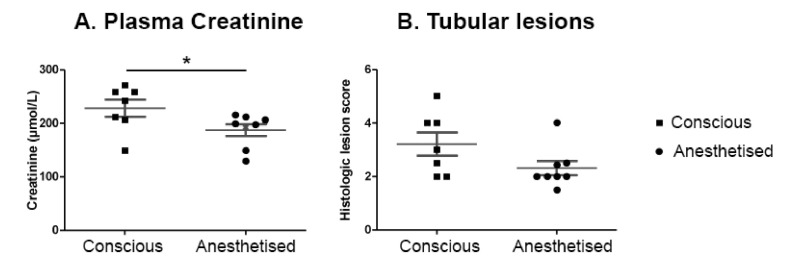
Anaesthesia ameliorated the outcome of a renal ischemia independently of the hypothermia. A 20-min renal ischemia in conscious animals induced a significant increase in plasma creatinine concentration (**A**) and a worsening of the renal lesions (**B**) compared to anaesthetised animals even when the body temperature is maintained at a similar level. Individual values are plotted, and the bars correspond to the average ± s.e.m. * *p* < 0.05 (Student’s *t*-test).

**Figure 8 ijms-22-09840-f008:**
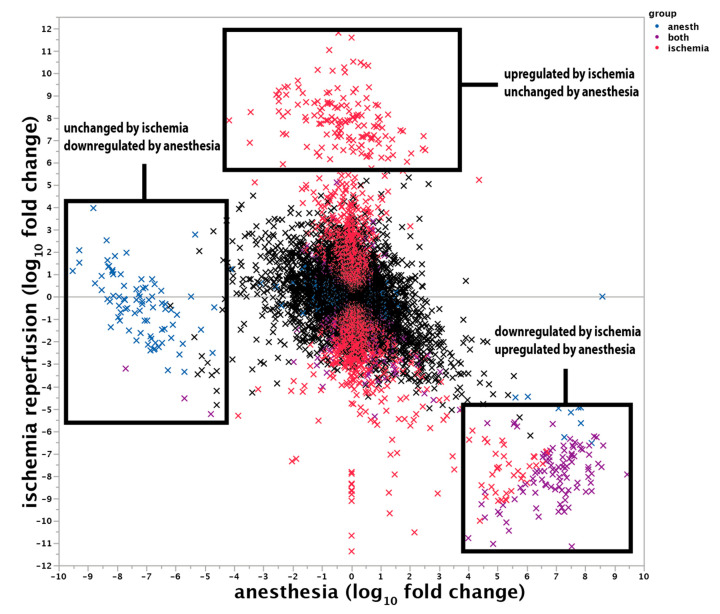
Respective effects of ischemia and anaesthesia on the renal transcriptome. The *y*-axis shows the fold change in the mRNA expression of ischemic kidneys compared to control kidneys, and the *x*-axis shows the fold change in the mRNA expression of ischemic kidneys under anaesthesia vs. ischemic kidneys in conscious mice. The statistical analysis is described in the Materials and Methods section.

## Data Availability

The data discussed in this publication have been deposited in NCBI’s Gene Expression Omnibus [33] and are accessible through GEO Series accession number GSE182793 (https://www.ncbi.nlm.nih.gov/geo/query/acc.cgi?acc=GSE182793), accessed on 28 August 2021.

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
