# Peer review of "Anaesthesia-Induced Transcriptomic Changes in the Context of Renal Ischemia Uncovered by the Use of a Novel Clamping Device"

_ijms, 2021, doi:10.3390/ijms22189840_

Round 1
Reviewer 1 Report
Authors have presented a valuable scientific work. It has a very important clinical significance.
Series of experiments have been designed properly.
Please find one more comment below:
Expression of KIM-1 mRNA was investigeted in the performed studies. There is no information about KIM-1 available in the introduction section hence this should be added. Furthermore, only the abbreviation can be found in the manuscript, so this should be corrected.
Otherwise, if this addition is done, the manuscript could be accepted.
Author Response
We would like to thank the Reviewer for his/her kind comments. As requested, we have added the following sentences to describe what KIM1 is:
In order to precisely the renal injury imposed by the open RIRI clamp and the ischemia when it is tightened, we measured the transcriptional expression of KIM1 (Kidney Injury Molecule 1) by RT-qPCR. KIM1 is type 1 transmembrane protein domain, the expression of which in the proximal tubule is highly increased upon aggression.
Reviewer 2 Report
In this paper, Verney et al demonstrate a novel method to induce renal ischemia which drastically reduces the need for anesthesia and thus proposes to permit the study of ischemia itself, separated from the effect of anesthesia.
This is a very interesting advance and the authors should be congratulated for their efforts, there are however a few points which need to be addressed to improve the work:
- Figure 2: the pictures are very small for the echo, extremely small for histology (especially if the reader is supposed to see “loss of the brush border, flattening of the tubular epithelium and intratubular casts”. This is also true for figure 4, histology is an important part of the demonstration and figures musts be adapted to demonstrate the points made by the authors.
- The PCR figures should be in log scale, and since the authors chose to use multiple houskeeping genes I’d recommend the Pfaffl technique rather than the now outdated 2ddCt. Moreover, the controls appear to be at zero, which doesn’t make sense if the authors used the 2ddCt method (for which the controls should be at 1)
- The statistics are extremely confusing. First, there is no paragraph in the methods to explain it, second the signs are all over the place, from one sign all the way up to 4. it makes the figure heavy and once you reach p<0,05, is it not enough to prove your point ? Why use multiple types of signs when you already have the brackets ? Are all the comparision in figure 5 really that relevant ? How come some figures have statistiques all over, while other have none ? Finally, whatever the amount of statistics, signs, bracket: nothing is explained in the figure legends about them. This is a key point to address.
- Also missing in all the figure legends: sampling time. How long after ischemia was the sample taken for histology, RNA, blood analysis, etc ?
- Figure 7 highlights a lot of work and it is somewhat a pity that not much more is done with it. Is it the goal of the authors to use this technique to conduct a thorough study of anesthesia protocols vs ischemia ? This figure could be further supplemented by imaging (such as done in supp fig 4) with a figure for each of the three conditions (ischemia only, anesthesia only, both).
Minor:
- According to the text, Supp figure 4 is supposed to be temperatures of different mice, not a gene map
Round 2
Reviewer 2 Report
This is the revision version of Verney et al’s demonstration of a novel method to induce renal ischemia which drastically reduces the need for anesthesia and thus proposes to permit the study of ischemia itself, separated from the effect of anesthesia.
The revisions are very satisfactory, and I only have minor remarks
- Figure 7: statistics are not stated in the legends
- Images are bigger, however I would still prefer larger images, as IJMS doesn’t have figure limitations
- Figures are clearer with the lighter use of statistics, however some remain very busy. Are all the displayed comparison necessary? Could the figures be less busy and thus easier to follow for the reader ?
Author Response
We would like to thank the Reviewer for her/his comments.
We now provide new enlarged versions of Figure 2, 4 and 6. We have also provided for the review process the original TIFF files, in which the figures are larger than in the manuscript.
For figure 5, in which there is indeed a lot of statistical information, we do believe that all comparisons are required. In order to facilitate the reading of the figure, we have attributed a colour to each experimental group and displayed the statistical information in grey.
For Figure 7, we have added the statistical used for the comparison.